# An Uncertainty-Distillation- and Voxel-Contrast-based Framework for One-shot Segmentation of Novel White Matter Tracts

**Hao Xu**[1]                                    HAXU0889@UNI.SYDNEY.EDU.AU

**Tengfei Xue**[1,2]                              TXUE4133@UNI.SYDNEY.EDU.AU

**Dongnan Liu**[1]                                DONGNAN.LIU@SYDNEY.EDU.AU

**Fan Zhang**[2,3]                                ZHANGFANMARK@GMAIL.COM

**Carl-Fredrik Westin**[2]                        WESTIN@BWH.HARVARD.EDU

**Ron Kikinis**[2]                                KIKINIS@BWH.HARVARD.EDU

**Lauren J. O'Donnell**[2]                        ODONNELL@BWH.HARVARD.EDU

**Weidong Cai**[1]                                TOM.CAI@SYDNEY.EDU.AU

[1] *University of Sydney, Australia*

[2] *Harvard Medical School, USA*

[3] *University of Electronic Science and Technology of China, China*

**Editors:** Accepted for publication at MIDL 2024

## Abstract

Diffusion-MRI-based white matter (WM) tract segmentation plays an important role in analyzing WM characteristics in healthy and diseased brains. The uncommon (novel) tract segmentation is important to the success of clinical brain operation and the reduction of postoperative complications. The massive WM tract annotations are time-consuming and need experienced neuroanatomists. Novel tract segmentation using only one annotated scan alleviates the above problems but is challenging. Existing fine-tuning-based studies achieve promising results but suffer from the feature overlap problem. In the work, we propose an uncertainty-distillation- and voxel-contrast-based one-shot novel WM tract segmentation framework, which includes an uncertainty distillation module to transfer semantic segmentation knowledge from base tracts to novel tracts and a voxel-wise multi-label contrastive module to adjust the feature embedding space so as to alleviate the feature overlap problem. We compare our method with several state-of-the-art (SOTA) methods that are designed to predict novel tract segmentation. The experimental results demonstrate that our method improves the one-shot segmentation accuracy of novel tracts in five experimental settings.

**Keywords:** Diffusion MRI, White Matter Tract Segmentation, Uncertainty-Distillation, Contrastive Learning.

## 1. Introduction

Diffusion Magnetic Resonance Imaging (dMRI) based WM tract segmentation is important in analyzing WM characteristics in healthy and diseased brains (Zhang et al., 2020, 2022). However, the current analysis is restricted by the number of tracts available from the segmentation. Although some common tracts (base tracts, e.g., corticospinal tract) have available annotations, many other uncommon tracts (novel tracts, e.g., superficial tracts)

do not have sufficient annotations in clinical practice. To reduce the use of tract annotations, fine-tuning-based semi-supervised studies transfer semantic knowledge from base tracts to novel tracts to improve few-shot novel tract segmentation accuracy (Lu et al., 2021; Liu et al., 2022; Lu et al., 2022). However, there may be a feature overlap problem (Song et al., 2023) between base tracts and novel tracts in the feature embedding space, which will reduce the segmentation performance of novel tracts. Single-label pixel-contrast semantic segmentation method (Wang et al., 2021) achieves promising results but cannot be introduced to WM tract segmentation, which is a multi-label voxel-wise classification task (a voxel can be classified into different tracts because tract fibers may cross or overlap together).

To this end, we propose an uncertainty-distillation- and voxel-contrast-based framework for one-shot novel tract segmentation (Figure 1), which includes an uncertainty distillation module and a voxel-wise multi-label contrastive module. In the uncertainty distillation module, the base tract segmentation knowledge is transferred from the teacher model to the student model. To alleviate the feature overlap problem, we design a multi-label voxel-contrast module that pulls or pushes the pair of voxels according to their label similarity to adjust the feature embedding space. We compare our proposed method with several SOTA methods on the HCP dataset. Experimental results in five experimental settings prove that our proposed method improves the one-shot segmentation accuracy of novel tracts.

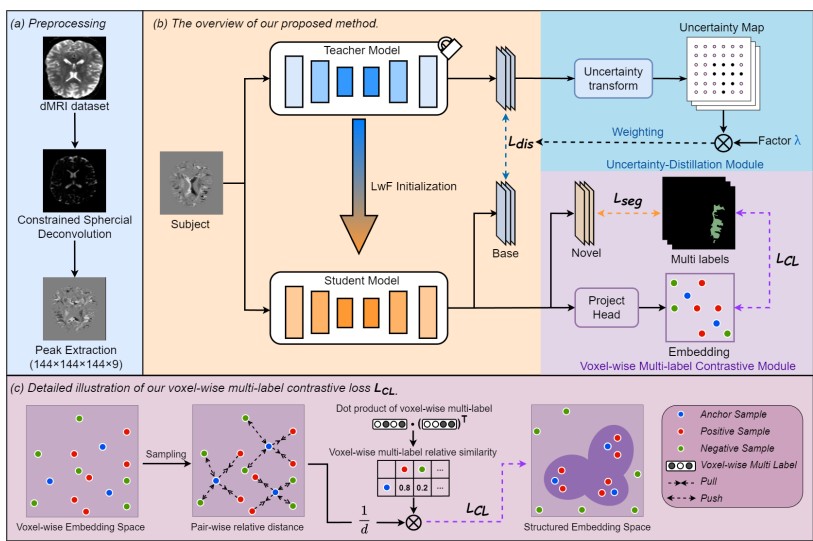

Figure 1: The overview of our proposed method.

## 2. Methods and Experiments

**Uncertainty Distillation Module.** First, the teacher model trained with only base tract labels is frozen and the LwF initialization (Li and Hoiem, 2017) is adopted to transfer the existing base tract segmentation knowledge into the student model. Second, uncertainty loss (Xu et al., 2023) is utilized to improve the distillation performance by filtering out voxels that are not trustworthy. The uncertainty distillation loss is the dot product of

the uncertainty map and the cross-entropy loss of the teacher model and base outputs of the student model. The segmentation loss is the cross-entropy loss of novel outputs of the student model and multi-labels.

**Voxel-wise Multi-label Contrastive Module.** We insert a project head into the student model for the voxel-wise feature embeddings and adjust the distance between the embedding of voxel pairs. Then, we pull the positive pairs closer and push the negative pairs further. We identify positive/negative pairs by normalized voxel-label similarity and normalized voxel-level feature distance. A voxel pair is defined as a negative pair if its feature distance is too close relative to the voxel label similarity, and as a positive pair if the feature distance is too far relative to the voxel label similarity. Set the voxel embedding be $z$ and its corresponding voxel-level multi-label be $\hat{y}$. Set $C_{pq} = \hat{y}_p^T \cdot \hat{y}_q$ is the label similarity of a voxel-pair and $g(p) = \{k | k \in 1, 2, ..., m + n, k \neq p\}$ is a set of voxels except for the voxel $p$. The voxel-wise multi-label contrastive loss $L_{CL}$ is as follow:

$$L_{CL}{}^{pq} = -\beta_{pq} \log \frac{e^{-d(z_p, z_q)/\tau}}{\sum_{k \in g(p)} e^{-d(z_p, z_k)/\tau}}, \tag{1}$$

where the dynamic coefficient $\beta_{pq} = \dfrac{C_{pq}}{\sum_{k \in g(p)} C_{pk}}$ is the normalization of $C_{pq}$, $d(\cdot, \cdot)$ is the Euclidean distance, and $\tau$ is a hyperparameter.

**WM Tract Dataset and Hyperparameters.** We use the dataset from (Wasserthal et al., 2018), which contains 85 subjects (base training: novel training: testing = 63:1:21) from the Human Connectome Project (HCP) (Van Essen et al., 2013). Each subject has 72 tract annotations, which been divided into base and novel tracts in the ratio of 5:1, 2:1, 1:1, 1:2, and 1:5. We use the multi-shell multi-tissue constrained spherical deconvolution (CSD) method (Tournier et al., 2007) with all gradient directions for transforming dMRI data to fiber orientation distribution function (fODF) peak data. Hyperparameter $\lambda$ is 1 and $\tau$ is 1.

**Quantitative Evaluation.** As shown in Table 1, the comparison and ablation experiments on five settings demonstrate the superiority of our proposed framework.

Table 1: Quantitative comparisons for one-shot novel tracts on the test set.

| | Method | Dice Score (%) | | | | |
|---|---|---|---|---|---|---|
| | | Base: Novel = 5:1 | Base: Novel = 2:1 | Base: Novel = 1:1 | Base: Novel = 1:2 | Base: Novel = 1:5 |
| Comparison | TractSeg(Wasserthal et al., 2018) | 0.48±0.76 | 3.16±11.9 | 1.86±9.55 | 1.87±8.95 | 1.40±7.57 |
| | VoxelMorph(Balakrishnan et al., 2019) | 59.43±7.45 | 59.22±6.40 | 59.52±5.70 | 57.75±8.46 | 59.17±8.19 |
| | CFT(Lu et al., 2022) | 63.90±14.68 | 54.70±19.28 | 58.22±18.84 | 50.37±23.70 | 51.29±27.12 |
| | IFT(Lu et al., 2022) | 77.21±4.51 | 67.42±15.55 | 59.55±19.54 | 24.11±18.73 | 14.81±18.80 |
| | TractSeg-LE(Liu et al., 2022) | 48.54±10.87 | 48.72±18.19 | 57.71±14.81 | 47.84±20.44 | 44.42±19.39 |
| Ablation Study | Seg + Dis | 77.68±5.18 | 74.13±5.58 | 72.33±7.98 | 69.73±17.64 | 69.26±17.73 |
| | Seg + Dis + CL (Ours) | **78.24±4.97** | **75.34±4.80** | **74.98±5.77** | **71.03±16.01** | **70.54±16.11** |

## 3. Conclusion

In this work, we propose an uncertainty-distillation- and voxel-contrast-based framework for one-shot tract segmentation. Comparison and ablation experiments in multiple experimental settings demonstrate the effectiveness of our proposed method. Our framework can be particularly valuable for studies with scarce tract labels (e.g., superficial white matter) or studies where high-quality annotations are difficult and expensive to obtain (e.g., high-resolution dMRI data).

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
