# OpenReview forum: "An Uncertainty-Distillation- and Voxel-Contrast-based Framework for One-shot Segmentation of Novel White Matter Tracts"
_MIDL.io/2024/Short_Papers — MIDL 2024 Short Papers_

### Official Review · Reviewer_dyUu · 2024-04-24

**Confidence:** 5
**Final Rating:** 5

**Review:**

This paper presents a new framework for one-shot segmentation of white matter tracts. This framework is composed of an uncertainty distillation module to transfer semantic segmentation knowledge from base tracts to novel tracts and a voxel-wise multi-label contrastive module.
Proposed method is compared with several SOTA methods on the HCP dataset showing its accuracy.
This method is very interesting and could be applied to other studies where high-quality annotations are difficult to obtain.

---

### Decision · Program_Chairs · 2024-04-26

Accept